Vertical assemblage of the holoplanktonic mollusks (Pteropoda and Pterotracheoidea: Carinaiidae, Pterotracheidae) in the Campeche Canyon, southern Gulf of Mexico, during a “Nortes” season

López-Cabello Zayra 1
Coria-Monter Erik coria@cmarl.unam.mx 2
Monreal-Gómez María Adela monreal@cmarl.unam.mx 2
Salas de León David Alberto 2
Durán-Campos Elizabeth 2
Gracia Adolfo 2
1 Facultad de Ciencias, Universidad Nacional Autónoma de México , Ciudad de México , Ciudad de México , Mexico
2 Instituto de Ciencias del Mar y Limnología, Universidad Nacional Autónoma de México , Ciudad de México , Ciudad de México , Mexico
Idris Izwandy
Electronic publication date: 2025 Mar 31
Publication date: 2025
Volume: 13
Electronic Location ID: e19118
Received 2024 Aug 23; Accepted 2025 Feb 14
Copyright: ©2025 López-Cabello et al.
Copyright year: 2025
Copyright holder: López-Cabello et al.
License: This is an open access article distributed under the terms of the Creative Commons Attribution License, which permits unrestricted use, distribution, reproduction and adaptation in any medium and for any purpose provided that it is properly attributed. For attribution, the original author(s), title, publication source (PeerJ) and either DOI or URL of the article must be cited.
License URL: https://creativecommons.org/licenses/by/4.0/

Keywords: Gastropods mollusks, Species diversity, Thermocline, Eddies, Gulf of Mexico, Zooplankton, Pterotracheoidea, Pteropoda

Funding: The Instituto de Ciencias del Mar y Limnología (UNAM) 144 145 627 DGAPA-PAPIIT-UNAM #IA200123 The UNAM CONAHCYT scholarship CVU914822 This study was funded by the Instituto de Ciencias del Mar y Limnología (UNAM), grants 144, 145, and 627, and DGAPA-PAPIIT-UNAM project #IA200123. The ship time for the CAÑON-IV expedition on board the R/V Justo Sierra was funded by the UNAM. Posgrado en Ciencias Biológicas (UNAM) sponsored Zayra López-Cabello’s postgraduate studies during this work with a CONAHCYT scholarship (CVU914822). The funders had no role in study design, data collection and analysis, decision to publish, or preparation of the manuscript.

==============================
This study examines the vertical assemblages of the holoplanktonic mollusks (Pteropoda and Pterotracheoidea) in the Campeche Canyon, southern Gulf of Mexico, during a “Nortes” season (February 21–28, 2011) and explores their relationship with the hydrography and the geostrophic circulation pattern. High-resolution hydrographic data were acquired during a multidisciplinary research cruise of 48 hydrographic stations. Zooplankton samples were collected at 24 stations from four depths (10, 50, 100, and 200 m) using a multiple open/closed net system. The results revealed a deep thermocline at a depth of 90 m and a circulation pattern dominated by cyclonic and anticyclonic eddies that induced cold and warm cores. Thirty-three Pteropoda and three Pterotracheoidea species were identified, with the highest richness at a depth of 100 m (just below the thermocline). The highest organism densities were observed at a depth of 10 m. The diversity index (H’) showed variations depending on the sampling depth, with the highest values (1.9 bits ind−1) at 100 and 200 m, while at 10 m depth the lowest values (1.45 bits ind−1) were observed. Multivariate analyses showed that dissolved oxygen, and temperature were the main environmental factors determining organism density.

Introduction

Marine zooplankton are one of the most diverse assemblages of animals on Earth, and they play a pivotal role in the ocean because of their position in trophic webs, providing food to numerous species of high economic and ecological value (Brierley, 2017). They also contribute to biological pumps owing to their ability to indirectly capture carbon in the pelagic zone and sink it to the bottom of the ocean (Le Quéré et al., 2005).

Holoplanktonic mollusks (Pteropoda and Pterotracheoidea (this last hereafter: heteropods)) are two independent key organisms in marine ecosystems because of their multiple roles. For example, owing to their position in the food web and diverse feeding habits (herbivores, omnivores, and carnivores), they prey on a wide variety of organisms (Wall-Palmer et al., 2016). They are attractive prey for organisms with the higher levels of trophic webs, including fish and marine turtles, because of their size (0.5–5 mm) (Moreno-Alcántara, Giraldo & Aceves-Medina, 2017).

Pteropods and heteropods, members of the phylum Mollusca, class Gastropoda, are organisms that have developed remarkable adaptations. These adaptations, honed over their evolutionary history, are crucial for their survival within the plankton throughout their life cycle (holoplankton) (Lalli & Gilmer, 1989). Pteropods, for instance, have a foot that has evolved into two paired swimming wings, a feature that significantly aids their survival. They belong to the subclass Heterobranchia, infraclass Euthyneura, order Pteropoda, which is further divided into the suborders Eutecosomata, Pseudotecosomata, and Gymnosomata (Bouchet et al., 2017). Heteropods, on the other hand, belong to the subclass Caenogastropoda, order Littorinimorpha, superfamily Pterotracheoidea, and include the families Atlantidae, Pterotracheidae, and Carinariidae (Bouchet et al., 2017); all of them have a fin-shaped foot that they use for swimming, with the ventral side of their bodies facing upwards.

Holoplanktonic mollusks ensure the transfer of energy and carbon throughout the food web, thereby contributing to the proper functioning of biological or carbon pumps (Peijnenburg et al., 2020). In addition, because of their thin shells of aragonite, the holoplanktonic mollusks are considered oceanographic proxies of ocean acidification and have recently been named “canary in the coal mine” (Oakes, Davis & Sessa, 2021), highlighting the imperative need for investigations of these groups.

In the Gulf of Mexico, four species of Pterotracheoidea were identified with a wide distribution: Cardiapoda placenta (Lesson, 1830), Pterotrachea hippocampus Philippi, 1836, P. coronata Forsskål, 1775 and Firoloida desmarestia Lesueur, 1817 (Suárez-Morales & Castellanos, 2009). In the southern gulf, some studies on Pteropoda reveal that two species: Creseis acicula (Rang, 1828), and Limacina trochiformis (A. d’Orbigny, 1835), constituted more than 90% of this group of organisms (Flores-Coto et al., 2013).

In the neritic epipelagic layer 14 species of Pterotracheoidea (Lemus-Santana et al., 2014b), and 27 Pteropoda (Lemus-Santana et al., 2014a) were identified in May and November. C. clava (Rang, 1828), C. virgula (Rang, 1828), L. inflata (A. d’Orbigny, 1834), H. inflatus (A. d’Orbigny, 1835) and L. trochiformis (A. d’Orbigny, 1835) that constitute more than 95% of the total abundance of Pteropoda (Lemus-Santana et al., 2014a).

During warmer season (May–June), four species, C. conica Eschscholtz, 1829, L. trochiformis, C. acicula, and Heliconoides inflatus (A. d’Orbigny, 1835) of the pteropod assemblages in the southern coastal and oceanic zones of the Gulf of Mexico constituted the maximum densities in the region and suggested that freshwater input is beneficial for this group of organisms (López-Arellanes et al., 2018).

All these studies provide very valuable information on both groups of organisms; however, the vast majority have mainly focused on (1) taxonomic composition, (2) shallow waters (<200 m depth), and (3) been mainly conducted in the warmest months (May to August) in the southern gulf. A region with a marked influence of freshwater input from the Coatzacoalcos River and the Grijalva-Usumacinta System, thus with a marked difference in terms of the composition of organisms in the oceanic region. Therefore, there are still gaps in the study of these organisms, first in oceanic waters within the Gulf and second in the coldest months.

This study aimed to analyze the vertical assemblages of holoplanktonic mollusks, Pteropoda, and Pterotracheoidea (Carinaiidae, Pterotracheidae) in Campeche Canyon, southern Gulf of Mexico, and to explore the role of hydrography and geostrophic circulation patterns on the composition, distribution, and density of these groups in the “Nortes” season (February 2011). We hypothesized that the hydrographic structure of the water column and the circulation pattern (characterized by a deep thermocline/pycnocline and the presence of cyclonic/anticyclonic eddies) would influence the composition distribution and density of both groups of organisms. We also aimed to contribute to the knowledge of these organisms in the oceanic region during the coldest season and identify their vertical distribution at specific depths.

Material and Methods

Study area

The Gulf of Mexico is the largest and deepest interior sea in North America, sharing waters with Mexico, the United States of America, and Cuba (Fig. 1A). The Campeche Canyon in the southern Mexican region of the Gulf is of tectonic origin. It is the deepest geomorphological feature in the region and reaches a depth of more than 3,500 m (Fig. 1B). The Campeche Canyon is highly dynamic and plays an essential role in the marine ecology of the region owing to the occurrence of different hydrodynamic processes, including internal waves (Santiago Arce & Salas de León, 2012), fronts (Aldeco-Ramírez et al., 2009), and eddies (Salas-de-León et al., 2004). These hydrodynamic processes determine the nutrient levels in the euphotic zone that are available for phytoplankton communities (Durán-Campos et al., 2017).

Figure 1 Study area: (A) The Gulf of Mexico; the rectangle in black shows the location of the Campeche Canyon. (B) The Campeche Canyon, bathymetry (m), hydrographic stations (+), zooplankton sampling stations (O).

T-T’ is a transect where the hydrographic properties of the water column and density values of organisms were analyzed.

The southern Gulf of Mexico is influenced by northerly winds from September to April (Salas-Monreal et al., 2022). It is called the “Nortes” season. Atmospheric forcing plays a critical role in this region. For example, during the “Nortes” season, extreme (>80 km h−1) and persistent northerly winds cross the gulf, exerting a significant influence on the hydrographic properties of the water column and inducing a deep mixed layer (>70 m depth) (Arriola-Pizano et al., 2022).

Studies on the composition, density, and distribution of zooplankton organisms in the southern Gulf of Mexico began in the middle of the last century because of investigations conducted by the Mexican government, particularly the Mexican Navy. Since then, the importance of holoplanktonic mollusks has been recognized because they are usually the dominant component after the copepods and chaetognaths (Toral Almazán et al., 2022). To date, the study of holoplanktonic mollusks in the Gulf of Mexico has evolved and has been approached in different ways, including taxonomic lists (e.g., Michel & Michel, 1991; Suárez, 1994), studies in neritic areas of the southern portion (e.g., Leal-Rodríguez, 1965; Matsubara-Oda, 1975; Flores-Coto et al., 2013; Lemus-Santana et al., 2014a; Lemus-Santana et al., 2014b), and studies that evaluated the influence of freshwater discharges (e.g., Lemus-Santana, 2011). Most of these studies include pteropods, while heteropods have only been considered in a few cases (e.g., Lemus-Santana et al., 2014b; Espinosa-Balvanera, 2017). Based on these investigations about 53 pteropods and 15 heteropods species were identified.

Sampling

The hydrographic dataset and zooplankton samples considered in this study were recorded during a multidisciplinary scientific expedition “CAÑON-IV” in the Campeche Canyon during February 21–28, 2011, on board of the R/V Justo Sierra owned by the Universidad Nacional Autónoma de México (UNAM). Hydrographic data were acquired at 48 stations equidistantly separated every 14 nautical miles (+ symbols in Fig. 1B: Table S1) using a conductivity-temperature-depth probe (CTD, SeaBird-19 plus) configured with oxygen and chlorophyll-a fluorescence sensors (SBE-43 and ECO-Wet Labs, respectively) which were calibrated pre- and post-cruise and placed in a Rosette (General Oceanics). Each CTD cast was taken from the sea surface to close to the bottom (5 m above the sea floor). The probe was lowered at a rate of 1 m s−1 with the interface configured to store data at 24 Hz.

Immediately following each Rosette-CTD cast, zooplankton samples were collected, during both daytime and nighttime, from a total of 24 stations (O symbols in Fig. 1B), at four depths (10, 50, 100, and 200 m), using a double trip mechanism close-open-close system (General Oceanics) and conical nets (75 cm diameter, 250 cm total length, and 505 µm mesh size) configured with a flowmeter (General Oceanics, model 2030R) placed in the mouth of each of the nets. A vertical array of four systems was placed on a hydrographic winch wire, and the cosine of the wire angle was calculated to ensure the capture of organisms at each depth (Kramer et al., 1972). Each net was opened by manual messengers starting the haul at 2 knots for 15 min; once the hauls finished, the nets were closed again with manual messengers and then recovered. Once on board, the organisms collected were fixed with formalin (added sodium borate) at 4% for 24 h. The final preservation was in 70% ethanol in glass jars with tight lids. The samples were stored under special precautions to avoid degradation of the organisms until analysis; for example, we usually changed both the airtight lids and the ethanol every two months.

Laboratory analyses

Holoplanktonic mollusks were obtained from 65 samples collected at 24 sampling stations in the Campeche Canyon. Complete and undamaged specimens of both groups were picked and separated in a glass Petri dish; approximately 20,000 specimens were sorted. They were identified at the species level using a Carl Zeiss Stemi 508 stereomicroscope equipped with a Zeiss Axiocam 105 color following the standard keys of Richter & Seapy (1999); Van der Spoel & Dadon (1999); Gasca & Janssen (2014), and Wall-Palmer et al. (2018). The identification was compared and confirmed using online open-access databases (e.g., MolluscaBase, Biodiversity Heritage Library, Tree of Life web project, and local repositories, such as the National Commission for the Knowledge and Use of Biodiversity (CONABIO, Spanish acronym)). The number of organisms was finally standardized to density units as number of individuals per 100 cubic meters (ind 100 m−3) using registers of flowmeters placed in the mouth of each net, following the standard protocols of Kramer et al. (1972), and Harris et al. (2000). After adult organisms’ separation, we realized taxonomic determination at the lowest possible level using specialized literature as Burridge et al. (2019), Gasca & Janssen (2014), Janssen, Bush & Bednarsek (2019), Richter & Seapy (1999), Tesch (1950); Van der Spoel, Bleeker & Kobayasi (1993). The main characteristics analyzed were the shell shapes and ornamentations (keels, notches, tubercles, reticulations, number of ribs and striation, and spines). For gymnosomes, the anatomy of the gills, pedal lobes, and oral structures (cones, arms, and suckers) were considered. The shape of the visceral nucleus was analyzed for heteropod mollusks of the Pterotracheidae and Carinariidae families, as they are organisms with sexual dimorphism, we analyzed the presence of reproductive structures such as gonads, tentacles, and suckers. The specimens were photographed from various angles using the manufacturer’s software (Zeiss ZEN Core) to visually document the species. Subsequently, the images were processed using Helicon Focus software version 8.2.0 and edited using Adobe Photoshop software version 23.5.1.

Data reduction

The CTD data were initially processed using the routines included in the software provided by the manufacturer (SBE Data Processing v.7.26.7). This involved removing poor-quality data and averaging the remaining data to 1 dbar. Then, conservative temperature (°C), absolute salinity (g kg−1), and density (sigma-t, kg m−3) were calculated using the algorithms of the thermodynamic equation for seawater, TEOS-10 (IOC, SCOR and IAPSO, 2010). These data were then used to construct vertical cumulative profiles to identify the thermocline/pycnocline depth, which was later confirmed using the depth of the maximum temperature vertical gradient (∂T/∂z). Geostrophic velocities relative to a depth of 1,000 m or to the bottom at shallower stations were calculated following the standard methods described by Pond & Pickard (1995). Finally, the vertical section shows the hydrographic parameter distributions constructed along transect T-T’ (Fig. 1B).

The dataset generated with the physical-biological parameters was organized in matrices that included the density value of each species at each sampling depth and the corresponding physical variables (e.g., temperature, salinity, and current speed). The organism density values at each sampling depth are represented using box-and-whisker plots. The relative abundance of each taxon was ordered in bar graphs stacked at 100% to analyze the variation in the composition and organism’s density values of holoplanktonic mollusks at each sampling depth (10, 50, 100, and 200 m). These graphs were realized with the “ggplot2” library (Wickham, 2016) in the R Studio software (R Core Team, 2021).

The diversity index (H’) at each sampling station for each sampling depth was calculated following Shannon & Wiener (1949) using the “biodiversityR” (Kindt & Coe, 2005) and “vegan” (Oksanen et al., 2022) libraries in R Studio software (R Core Team, 2021). These values were represented in graphs for each sampling depth using box-and-whisker plots built with the “ggplot2” library (Wickham, 2016) in the R Studio software (R Core Team, 2021). The Kruskal-Wallis test was used to analyze the significance of the differences in density, species richness, and diversity about depth.

Finally, principal component analysis (PCA) was conducted to analyze the physical variables affecting holoplanktonic mollusk assemblages at each sampling depth. Additionally, to identify similarities between the holoplanktonic mollusk communities during sampling, non-metric multidimensional scaling (NMDS) and SIMPROF tests were applied. These analyses were performed following the standard routines of the Plymouth Routines in the Multivariate Ecological Research (PRIMER, v7) software (Clarke, 1993). Finally, we realized a Pearson correlation for to analyze the statistical significance between density of each dominant species and environmental variables included in the PCA.

Results

Hydrography

The vertical distribution of the hydrographic parameters in the upper 200 m shown in cumulative profiles indicated that the mean profiles of conservative temperature ranged from 24 to 14 °C, with the maximum vertical gradient observed at 90 m depth (Fig. 2A). Absolute salinity values varied from 35.8 g kg−1 at the surface and increasing to 36.75 g kg−1 at 70 m depth, then decreasing to 36.0 g kg−1 (Fig. 2B). Density (sigma-t) varied from 24.1 to 26.7 kg m−3 (Fig. 2C). The thermocline/pycnocline depth during sampling period was at 90 m (Fig. 2). This was estimated by the maximum vertical temperature gradient (∂T/∂z).

Figure 2 Cumulative vertical profiles of the hydrographic parameters in the upper 200 m layer in the Campeche Canyon in a “Nortes” season (February 2011): (A) Conservative temperature (° C), (B) absolute salinity (g kg−1) and (C) density (sigma-t, kg m−3).

The solid black line represents the mean profile.

The horizontal distribution of hydrographic parameters at a thermocline/pycnocline depth (90 m) showed features of warm and cold cores in the study area. The conservative temperature ranged from 18.7 to 21.5 °C displaying a large cold tongue in the southern region, a warmer northern region, and the presence of two fragmented cores with different temperatures in the central portion (Fig. 3). The absolute salinity ranged from 36.5 to 36.7 g kg−1 (see Fig. S1). The presence of a northern region with relatively higher salinity, as well as a small tongue of lower salinity in the southern region were observed. However, in general it was very uniform along the domain. The density values ranged from 25.8 to 26.2 kg m−3; as expected, their distribution displayed a similar pattern to those of the conservative temperature, with cores of different density values (Fig. 4). The circulation pattern showed westward geostrophic currents in the northern portion and southward currents in the eastern portion, reaching values of approximately 30 cm s−1. The cyclonic circulation in the southern region coincides with the tongue of low temperatures and salinity. Additionally, the presence of two small eddies, one cyclonic eddy (centered at 20.9°N and 93.2°W) and an anticyclonic eddy (centered at 20.3°N and 92.6°W) (Fig. 4), agree with the high and low densities, respectively. In the southwestern study area, horizontal density distribution at 90 m depth, at the beginning of the TT’ transect (Sta. 37; see Fig. 1B), shows a pair of cores low-density and high density, this last with a cyclonic geostrophic circulation. Generally, on the boundary between them there is a thermohaline front. On the other hand, at station 17 (see Fig. 1B) a high-density core is observed (Fig. 4).

Figure 3 Horizontal distribution of the conservative temperature (°C) at the thermocline depth (90 m).

Black points, sampling stations where the depth was <90 m.

Figure 4 Horizontal distribution of density (sigma-t, kg m−3), and geostrophic circulation pattern (cm s−1) at the thermocline depth (90 m).

Black points, sampling stations where the depth was <90 m.

The vertical distribution of the conservative temperature, absolute salinity, and sigma-t confirmed that the thermocline, halocline, and pycnocline were at a depth of 90 m with variations along the transect (Figs. 5A–5C). On the other hand, at station 17 (see Fig. 1B), from 200 m to approximately 80 m depth, the uplift of isolines suggests the occurrence of the cyclonic eddy (Figs. 5A–5C). In the upper 40 m layer, the variables show notable changes close to Sta. 37, indicating the front effect.

Figure 5 Vertical distribution along the transect T-T’ of (A) Conservative temperature (°C), (B) absolute salinity (g kg−1), and (C) density (sigma-t; kg m−3).

Holoplanktonic mollusks assemblages

From a total of 19,964 organisms recorded, only 16,235 were identified at the species level belonging to two orders, three suborders, 15 families, 27 genera, and 36 species. Among them, 33 were order Pteropoda, of which four (Pneumodermopsis macrochira Meisenheimer, 1905, Spongiobranchaea intermedia Pruvot-Fol, 1926, Schizobrachium cf. polycotylum Meisenheimer, 1903, and Cliopsis krohnii Troschel, 1854) were recently identified, in a collateral study, as new records for the southern Gulf of Mexico (López-Cabello et al., 2024). Some specimens identified in this study are shown in Fig. 2. Three species were of the superfamily Pterotracheoidea (Carinaiidae, Pterotracheidae) (Table 1). The pteropods Heliconoides inflatus, Creseis conica, and Limacina trochiformis presented the highest density values (2,228.5, 1,541.7, and 1,416.7 ind 100 m−3, respectively), representing 69.1% of the total density. In contrast, the pteropods Clio cuspidata, Cymbulia peronii, Clione limacina, and the heteropod Pterotrachea hippocampus presented values of <0.3 ind 100 m−3, the lowest among all samples (Table 1). C. conica is dominant at 10 m depth, while H. inflatus is dominant at 50, 100 and 200 m when comparing the density of the species at different depths. The species’ dominance at different depths is such that at 10 m, C. conica has the highest density, followed by H. inflatus and finally L. trochiformis. At 50 m, the species with the highest density is H. inflatus, followed by C. conica and finally L. trochiformis. At 100 and 200 m, the hierarchy in terms of dominance is H. inflatus, L. trochiformis, and C. conica. Considering the total density of each species at each of the four depths, the dominant species is H. inflatus, followed by C. conica, and L. trochiformis (Table 1).

Table 1 Holoplanktonic mollusks (Pteropoda and Pterotracheoidea) and their density values (ind 100 m−3) in the Campeche Canyon (southern Gulf of Mexico) in a “Nortes” season (February 2011), at each sampling depth.

Bold values represent the three dominant species.

Order Pteropoda	Density (ind 100 m−3)	
	10 m	50 m	100 m	200 m	Total	
Cavolinia uncinata (A. d’Orbigny, 1835)	7.0	20.5	12.2	5.2	44.8	
Cavolinia inflexa (Lesueur, 1813)	246.2	162.6	89.0	50.6	548.5	
Diacavolinia limbata (A. d’Orbigny, 1835)	8.4	0.9	0.0	0.0	9.3	
Diacavolinia stangulata (Deshayes, 1823)	12.0	4.1	1.4	0.0	17.5	
Diacavolinia constricta van der Spoel, Bleeker & Kobayasi, 1993	8.5	0.0	0.4	0.4	9.4	
Diacavolinia longirostris (Blainville, 1821)	1.2	0.6	0.0	0.0	1.8	
Diacria major (Boas, 1886)	0.0	0.3	1.2	0.0	1.5	
Telodiacria danae (van Leyen & van der Spoel, 1982)	98.0	58.1	18.2	8.3	182.6	
Clio pyramidata Linnaeus, 1767	0.0	0.0	1.8	0.0	1.8	
Clio cuspidata (Bosc, 1801)	0.0	0.0	0.3	0.0	0.3	
Creseis acicula (Rang, 1828)	237.7	109.6	13.1	69.7	430.1	
Creseis conica Eschscholtz, 1829	830.2	403.1	214.1	94.3	1541.7	
Creseis virgula (Rang, 1828)	2.7	1.8	0.0	0.0	4.5	
Boasia chierchia e (Boas, 1886)	0.7	0.0	0.0	0.0	0.7	
Styliola subula (Quoy & Gaimard, 1827)	22.5	15.0	7.4	0.9	45.9	
Hyalocylis striata (Rang, 1828)	74.2	58.0	104.1	34.6	270.9	
Heliconoides inflatus (A. d’Orbigny, 1835)	683.5	991.3	367.3	186.5	2228.5	
Limacina bulimoides (A. d’Orbigny, 1835)	4.1	8.0	7.6	0.9	20.6	
Limacina lesueurii (A. d’Orbigny, 1836)	50.6	159.3	27.1	3.9	240.9	
Limacina trochiformis (A. d’Orbigny, 1835)	679.0	383.1	247.0	107.7	1416.7	
Peracle diversa (Monterosato, 1875)	0.0	0.0	0.7	0.3	1.0	
Peracle reticulata (A. d’Orbigny, 1835)	4.4	9.9	15.1	2.1	31.5	
Peracle bispinosa Pelseneer, 1888	0.0	0.0	0.0	1.9	1.9	
Desmopterus papilio Chun, 1889	17.8	36.4	122.0	46.9	223.2	
Cymbulia peronii Blainville, 1818	0.0	0.0	0.3	0.0	0.3	
Clione limacina (Phipps, 1774)	0.0	0.1	0.0	0.0	0.1	
Paraclione longicaudata (Souleyet, 1852)	8.4	0.9	2.0	0.0	11.4	
Thliptodon diaphanus (Meisenheimer, 1902)	0.0	4.1	9.0	2.1	15.2	
Cliopsis krohnii Troschel, 1854	0.0	1.4	0.3	0.3	2.0	
Pneumoderma violaceum A. d’Orbigny, 1835	7.2	7.7	6.3	7.8	29.0	
Pneumodermopsis macrochira Meisenheimer, 1905	0.0	0.0	0.0	0.6	0.6	
Schizobrachium polycotylum Meisenheimer, 1903	0.7	0.3	0.2	0.0	1.2	
Spongiobranchaea intermedia Pruvot-Fol, 1926	2.2	2.5	0.0	0.0	4.7	
Superfamily Pterotracheoidea (Carinaiidae, Pterotracheidae)	
Carinaria pseudorugosa Vayssière, 1904	2.9	0.0	3.5	0.0	6.4	
Firoloida desmarestia Lesueur, 1817	18.2	80.6	42.8	11.4	153.1	
Pterotrachea hippocampus R. A. Philippi, 1836	0.0	0.0	0.2	0.0	0.2	

The density, species richness, and diversity of total species differ depending on the depth of the sampling. These differences in depth are significant according to the Kruskal-Wallis test, showing p < 0.05. For example, the highest density values were found at 10 m depth (median = 127.2 ind 100 m−3, mean = 205.7 ind 100 m−3), followed by the 50 m depth level (median = 68.2 ind 100 m−3, mean = 161.6 ind 100 m−3), whereas the 100 m and 200 m levels presented lower density values (Fig. 6A). The highest species richness was recorded at a depth of 100 m (28 species), whereas the lowest richness value was observed at a depth of 200 m (21 species). At depths of 10 and 50 m, the richness was 25 and 26 species, respectively (Fig. 6B). The diversity values (H’) also showed variations depending on the sampling depth, with the highest value at 100 and 200 m depth (>1.9 bits ind−1), while 10 m depth presented the lowest value (<1.5 bits ind−1) (Fig. 6C). The vertical distribution of the species that showed the highest density values was explored along T-T’ transect (see Fig. 1B) in relation to the hydrographic parameters. The vertical distribution of H. inflatus along this transect showed that the values varied in a range of 1.3 to 255.2 ind 100 m−3, with the maximum one observed at 10 m depth in the station 37. The second-highest relative value was observed at a depth of 200 m at Station 35 (Fig. 7A). C. conica ranged from 0.4 to 255.7 ind 100 m−3, with a similar vertical distribution pattern recording the highest values at 10 m depth in station 37 (Fig. 7B). L. trochiformis values ranged from 0.4 to 162 ind 100 m−3 (Fig. 7C), showing two maximum peaks located in the surface waters (10 m depth), the maximum in station 17 and a second peak in station 37.

Figure 6 Holoplanktonic mollusks in the Campeche Canyon in a “Nortes” season (February 2011) at each sampling depth: (A) Organism density values (ind 100 m−3), (B) species richness, and (C) diversity (bits ind−1).

The dots represent the outliers.

Figure 7 The organism’s density values (ind 100 m−3) of the three dominant species superimposed over the vertical section water density distribution (kg m−3), (A) Heliconoides inflatus (A. d’Orbigny, 1835), (B) Creseis conica Eschscholtz, 1829, and (C) Limacina trochiformis (A. d’Orbigny, 1835).

The radius of circles is proportional to the density value of each species.

Statistical analyses

The PCA analysis showed that the two first axes explained 53.1% of the accumulated variance. In the first principal component (PC1), dissolved oxygen, conservative temperature, and absolute salinity were the variables with the highest influence, whereas in the second principal component (PC2), longitude had the highest influence (Table 2). Pearson correlation analysis between the density of each dominant species and environmental variables registered in this study showed significant correlation (p < 0.05), between Heliconoides inflatus and longitude, between Lamicina Trochiformis and sampling depth, temperature, and dissolved oxygen and, between Creseis conica and sampling depth, and temperature.

Table 2 Eigenvectors (coefficients in the linear combinations of variables: Latitude, Longitude, absolute salinity, conservative temperature, fluorescence of chlorophyll-a, dissolved oxygen, geostrophic velocity speed, and sampling time).

Values in bold show the variables that contributed most to each axis.

Variable	PC1	PC2	
Latitude	−0.313	−0.383	
Longitude	−0.016	0.668	
Absolute salinity	−0.423	0.180	
Conservative temperature	−0.443	0.194	
Fluorescence of chlorophyll-a	−0.326	0.351	
Dissolved oxygen	−0.518	−0.270	
Gesotrophic velocity speed	−0.383	0.001	
Sampling time	−0.072	−0.376	

The PCA diagram (Fig. 8A) shows that the data obtained at a depth of 200 m were opposite to chlorophyll-a fluorescence, conservative temperature, absolute salinity, and geostrophic speed. The opposite pattern was observed in the data obtained at a depth of 10 m, where dissolved oxygen, conservative temperature, absolute salinity, and chlorophyll-a fluorescence strongly influenced stations 1, 3, 5, 17, 29, and 31. The PCA diagram also shows that some stations at different depths (31, 39, 41, and 42) in the southeastern region presented a relationship with longitude.

Figure 8 (A) Principal component analysis diagram, and (B) non-metric multidimensional scaling applied to our data set.

Abbreviations are Lon, Longitude; F, Fluorescence of chlorophyll-a; T, Conservative temperature; S, Absolute salinity; V, geostrophic current speed; DO, dissolved oxygen; Lat, Latitude; ST, Sampling time.

The NMDS (with a stress of 0.13) and SIMPROF analyses (Fig. 8B) showed seven groups, the first two corresponding mainly to the stations at a depth of 10 m. Two additional groups were included at the depth of 50 m. The fifth and sixth groups included stations at 100 m depth, whereas the seventh group included stations at 200 m depth. This suggests that the composition and density of the holoplanktonic community in Campeche Canyon, at the time of our observations, presented differences depending on the depth at which they were collected.

Discussion

The taxonomic list obtained in our study included 33 pteropods and three heteropods, which seems lower than previous reports that documented up to 51 pteropods (e.g., Suárez, 1994). However, it is important to note that the method used in our study differs significantly; while other authors predominantly used Bongo net hauls to collect organisms from depths ranging from 200 m to the surface, other studies employed a multiple close-open-close net system, sampling layers at different depths, thus covering a more significant water column sample. In contrast, our study collected organisms from four specific depths, which could explain the differences in the number of species. Another difference is that our study was conducted in oceanic waters, whereas other investigations were conducted on the continental shelf, naturally affecting the number of collected species and densities.

The thermocline is a barrier that concentrates nutrients and planktonic organisms and has been documented in coastal (Durán-Campos et al., 2019) and oceanic environments (Signoret et al., 2006). In the southern Gulf of Mexico, there are seasonal changes in the thermocline depth. The depth is shallow (40 m depth) in summer (Salas-de León et al., 2004) and deeper (90 m depth) in autumn and winter (Durán-Campos et al., 2017; Arriola-Pizano et al., 2022). We confirmed a deep thermocline in a “Nortes” season (February 2011). This seems to affect the richness and diversity of holoplanktonic mollusks of this study, since the maximum diversity was at 100 and 200 m, while the maximum richness was found at 100 m depth, just below the thermocline. These types of zooplankton aggregates have been documented around the thermocline in different areas of the planet. For example, in the Andaman Sea (northeastern Indian Ocean), the vertical distribution of zooplankton, including holoplanktonic mollusks, tends to be higher at the thermocline (Madhupratap et al., 1981). In the Gulf of Patraikos (Ionian Sea, Greece), most zooplankton groups tended to aggregate in the thermocline layer (Fragopoulu & Lykakis, 1990). More recently, it was documented that zooplanktonic organisms are restricted above or below the thermocline in the Northern Yellow Sea (Western Pacific Ocean) (Ge et al., 2021). In our study, the maximum diversity was found at 100 and 200 m depth. The maximum richness was at 100 m, close to the thermocline. This could be related to the vertical distribution of phytoplankton biomass in the region. Recent evidence documented that in the Campeche Canyon in a “Nortes” season (February 2011), the chlorophyll-a maxima were deep, and were located between 90 and 100 m depth (Torres-Martínez et al., 2020). This may increase food availability for the holoplankton mollusks.

Although zooplankton have developed several migration strategies throughout the water column, numerous studies have indicated that their distribution, composition, and abundance in oceans depend on the presence of several hydrodynamic processes at different spatial and temporal scales, such as internal waves, fronts, upwelling, and eddies, which are linked to energy transport and nutrient supply to the euphotic layer (McGillicuddy Jr, 2016). The influence of physical processes on the composition and distribution of zooplankton in Mexican waters has been documented. For example, in the Bay of La Paz, southern Gulf of California, recent evidence suggests that the circulation pattern, dominated by the presence of a cyclonic eddy, exerts a notable influence on the zooplanktonic organisms. This generates a differential distribution from the center to the periphery of the eddy and forms a circular arrangement that has been called a “copepod belt” (Rocha-Díaz et al., 2021). In the southern Gulf of Mexico, particularly in the Campeche Canyon, García-Álvarez (2015) analyzed the vertical structure of ichthyoplankton organisms during summer and observed that high organism densities were influenced by a shallow thermocline (30 m depth). In addition, it was also observed that a circulation pattern was dominated by the presence of a cyclonic eddy, as was in our case. More recently, Arriola-Pizano et al. (2022) analyzed the distribution, density, and diversity of euphausiids and their relationship with the hydrodynamic processes in different climatic seasons (at the end of summer and “Nortes”), showing that a frontal region between a cyclonic/anticyclonic eddy established the ideal temperature conditions to host high values of richness and organism abundance. The authors also noted that Campeche Canyon has heterogeneous habitats because of the hydrodynamic processes that benefit zooplanktonic organisms. Furthermore, they also highlighted that during the “Nortes” season, the thermocline reached a depth of 80 m (as was in this case), affecting the distribution of organisms.

Unlike the coastal zone, where C. acicula is the dominant species in August (Flores-Coto et al., 2013); and in May (Lemus-Santana et al., 2014a), and other authors considered C. conica as a dominant species in May–June (López-Arellanes et al., 2018), in the oceanic zone (Campeche Canyon), H. inflatus is the dominant species in February, “Nortes” season. The PCA applied to our dataset showed that several notable environmental variables influenced the distribution of the identified species, depending on the sampling depth. For instance, the diagram (Fig. 8A) suggests that the dissolved oxygen was the first factor explaining organism density variation in the first component. It has been documented that this variable affects mollusks in general (Kuk-Dzul & Díaz-Castañeda, 2016), and in particular pteropods, specially stimulating high density values of species of the Limacina genus (Engström-Öst et al., 2019), as was our case.

The temperature was the second physical factor affecting the organism density in the first two levels (10 and 50 m depth), which is an essential variable related to the metabolism of holoplanktonic mollusks (López-Arellanes et al., 2018). As mentioned above, both pteropods and heteropods are particularly sensitive to environmental changes. Dissolved oxygen and temperature are among the variables that exert a significant effect, which usually affects both groups’ metabolism (respiration and excretion). Notably, in pteropods, the highest respiration and, usually, highest excretion rates have been reported to occur at higher temperatures (Thibodeau, Steinberg & Maas, 2020), while low temperature tends to suppress the respiration of pteropods by ∼80–90% (Maas, Wishner & Seibel, 2012). Furthermore, changes in the temperature regime affect feeding behavior and the morphology of pteropods’ feeding structures (Seibel, Dymowska & Rosenthal, 2007), which could potentially impact their survival. In order of importance, salinity was another physical factor which seems to affect the organisms of interest in this study (Table 2). Recent studies in the central Mediterranean Sea on pteropod populations showed that the most abundant C. conica reported a negative correlation with salinity (Béjard et al., 2024; Johnson, Manno & Ziveri, 2023), but only very low salinities have been shown to affect pteropods distribution negatively (Béjard et al., 2024). In our case the salinity range was within the normal range reported for pteropods and its variation range very narrow (1.5 g kg−1). Even though the interaction of increasing salinity and temperature has been reported to increase the biomass of L. helicina in the Siberian Arctic Seas at a small salinity variation range (Pasternak et al., 2020). The effect of salinity on the abundance of pteropods was controversial. In the southern Gulf of Mexico, in low-salinity coastal waters, the abundance of L. trochiformis was inversely correlated with salinity (López-Arellanes et al., 2018), whereas a similar negative correlation of C. conica abundance was observed in the central Mediterranean with high salinities (Béjard et al., 2024). Both species are among the most abundant in Campeche canyon but were found in a narrow salinity (from 35.5 to 37.0 g kg−1) which is the common range for pteropods distribution.

The geostrophic velocity shows two features: front (the boundary between currents) and cyclonic eddies, which promotes high productivity. The front propitiates an accumulation of detritus that increases productivity. The principal mechanism in the cyclonic eddy is the uplift of the nutricline, which induces nutrient input into the euphotic layer (McGillicuddy Jr, 2016). Thus, both features favor a high density of holoplanktonic mollusks.

In addition, the longitude in the second principal component longitude seems to affect the distribution (in this case, horizontal) of the holoplanktonic mollusks, which explains the important spatial variations in the community structure in this study.

Knowledge of the composition, density, and distribution of holoplanktonic mollusks and their relationship with physical processes in Campeche Canyon is still scarce. However, some studies conducted in the shallow waters of the southern continental shelf of the Gulf of Mexico (e.g., Flores-Coto et al., 2013; López-Arellanes et al., 2018) have identified that the physical environment, particularly the temperature regime, current pattern, and freshwater discharge, play a fundamental role in community structure. Unfortunately, these studies have primarily been conducted during the warm season, leaving gaps and uncertainties regarding the response of holoplanktonic mollusks to the oceanic dynamics that occur during the cold season. This study represents the first attempt to advance knowledge in this area. However, more research is needed to cover a complete annual cycle and to evaluate interannual variability. For instance, the impact of large-scale events such as the El Niño-Southern Oscillation and North Atlantic Oscillation (NAO), on the community structure of holoplanktonic mollusks in Campeche Canyon remains unknown. Therefore, it is necessary to implement long-term monitoring programs with the support of academia and governments to gain a comprehensive understanding of the system, thus enabling the proposal of effective actions to preserve the Gulf of Mexico’s highly biodiverse ecosystem.

Conclusion

The deep thermocline significantly impacted the richness and diversity of holoplanktonic mollusks in Campeche Canyon during the “Nortes” season (February 2011). The highest value of richness was observed at 100 m depth, just below the thermocline. The maximum diversity value was found at 100 and 200 m. This pattern may be linked to the vertical distribution of phytoplankton biomass. Three species, H. inflatus, C. conica, and L. trochiformis, had the highest organism densities. Statistical results indicated that several noteworthy environmental variables influenced the distribution of the identified species depending on the sampling depth. Temperature is one of environmental variables after dissolved oxygen affecting organism density values at the first two sampling levels, followed by salinity, which varied from 35.5 to 37.0 g kg−1 in the upper 50 m layer. The NMDS and SIMPROF analyses showed seven groups, with the first two corresponding mainly to stations at a depth of 10 m. Two additional groups were included at the depth of 50 m. The fifth and sixth groups included stations at 100 m depth, whereas the seventh group involved 200 m depth. Therefore, the results suggest that the composition and density of the holoplanktonic community in the Campeche Canyon in a “Nortes” season exhibited significant differences depending on the depth.

Supplemental Information

Supplemental Information 1 Horizontal distribution of absolute salinity (g kg−1) at the thermocline depth (90 m). Black points, sampling stations where the depth was <90 m

Supplemental Information 2 Species of the Order Pteropoda

a) Heliconoides inflatus (A. d’Orbigny, 1835), b) Creseis conica Eschscholtz, 1829, c) Limacina trochiformis (A. d’Orbigny, 1835), d) Cavolinia inflexa (Lesueur, 1813), e) Creseis acicula (Rang, 1828); y Superfamily Pterotracheoidea (Carinariidade, Pterotrachidae): f) Hyalocylis striata (Rang, 1828), g) Firoloida desmarestia Lesueur, 1817, h) Carinaria pseudorugosa Vayssière, 1904, In the Campeche Canyon during ”Nortes” season.

Supplemental Information 3 Species density

Supplemental Information 4 Environmental data

Supplemental Information 5 Sampling zooplankton station, total depth (m), date (m/d/y), local time (h), latitude, and longitude

We appreciate the assistance of the captain and his crew during the sampling expedition and students participating in the cruise. Octavio Quintanar-Retama, Sergio Castillo Sandoval and Francisco Ponce Núñez provided technical support during the analyses, we thank Benjamín Quiroz for his support in statistics, Jorge Castro improved the figures. We are grateful to the three reviewers whose comments and suggestions greatly improved our manuscript.

Additional Information and Declarations

Competing Interests

Author Contributions

Data Availability

The authors declare there are no competing interests.

Zayra López-Cabello conceived and designed the experiments, analyzed the data, prepared figures and/or tables, authored or reviewed drafts of the article, and approved the final draft.

Erik Coria-Monter conceived and designed the experiments, performed the experiments, analyzed the data, prepared figures and/or tables, authored or reviewed drafts of the article, and approved the final draft.

María Adela Monreal-Gómez conceived and designed the experiments, performed the experiments, analyzed the data, prepared figures and/or tables, authored or reviewed drafts of the article, and approved the final draft.

David Alberto Salas de León performed the experiments, authored or reviewed drafts of the article, and approved the final draft.

Elizabeth Durán-Campos conceived and designed the experiments, performed the experiments, analyzed the data, authored or reviewed drafts of the article, and approved the final draft.

Adolfo Gracia analyzed the data, authored or reviewed drafts of the article, and approved the final draft.

The following information was supplied regarding data availability:

The data is available at Zenodo: Monreal Gómez, M. A. (2025). Dataset article: Vertical assemblage of the holoplanktonic mollusks (Pteropoda and Pterotracheoidea: Carinaiidae, Pterotracheidae) in the Campeche Canyon, southern Gulf of Mexico, during a “Nortes” season [Data set]. Zenodo. https://doi.org/10.5281/zenodo.14602035.

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
