# Peer review of "Vertical assemblage of the holoplanktonic mollusks (Pteropoda and Pterotracheoidea: Carinaiidae, Pterotracheidae) in the Campeche Canyon, southern Gulf of Mexico, during a “Nortes” season"

_PeerJ, doi:10.7717/peerj.19118_

## Round 0.1 · original submission · Major Revisions

Overall, the data and outcomes from this study are suitable for publication. However, several issues highlighted by reviewers must be addressed before this manuscript moves to the next stage of publication.

As the academic editor for this manuscript, I decided to seek three reviews before making a decision. This is because one of the reviewers gave unusual comments (wrong line number, etc.). Thus, in your response letter, please mention which comments you have avoided and the reasons behind them.

It would be great if the revised version is proofread before submission.

Reviewer 1 ·

Basic reporting

I have reviewed this manuscript by Lopez-Cabello et al #103528 aims to examine the vertical assemblage of the holoplanktonic mollusks (Pteropoda and Pterotrachoidea) from 4 different depths in the Campeche Canyon, southern Gulf of Mexico during a ‘Nortes’ season". This manuscript is clearly written and effectively delivers its findings. However, some part required further elaboration.

Experimental design

The assessment of species identification is inadequately presented. Information on how the species was examined, such as their morphological characteristics and life stage for identification (e.g., adult or juvenile), should be provided. Was there any variation with depth in term of the life stages?

The time of sampling should be included in the methodology. Was the sampling conducted during the daytime, or was it continuous?

Fig 5.It should be improved by adding a box plot of the (a)..., and please also indicate the dots shown in the graph.

Table 2. Please revise the caption appropriately. Clarify what the bold font represents.

Validity of the findings

Results are clearly stated. However, it is recommended to add picture of dominant species in the graph e.g Fig. 7, and others in the supplementary

Line 374 mentions the effects of temperature. This point should be further elaborated. How this could be related to the metabolism of holoplanktonic mollusks". Similarly, for geostrophic current speed, also required further elaboration.

The conclusions are well presented and align with the research objectives

·

Basic reporting

The article contains updated taxonomic names, as well as a well-structured phylogeny of the animals under study (e.g., the use of Pteropoda and Pterotracheoidea= Heteropoda).
It has defined introduction: first, the authors make an approach to planktonic mollusks in general. Then, they focuse on those from the Gulf of Mexico and finally address those who occupy the present study
About the methodology, the authors made an exhaustive description of the sampling, materials used, fixation and conservation methods. The bibliography used for species identification is varied, valid and up-to-date. The data analysis is coherent, rigorous and with a correct use of algorithms and software.

CORRECTIONS:

I can’t see the description of the figures (figure caption). If they don't exist, they should be added.
In keywords section, I would add some of them more: zooplankton; pterotracheoidea; pteropoda
In the second paragraph of Introduction (lines 51 and therefore), I would start by talking about basic characters and phylogeny of the group (only two or three sentences). I recommend reviewing and including in the bibliography the classic work of Lalli & Gilmer, 1989, “The biology of holoplanktonic gastropod mollusks”.

In line 63, where says “some studies on heteropods reveal that”…” looking at the species named after this, authors may refers to Pteropods, not heteropods.
In the same line (63) where says “reveal that three species of Pteropoda”, there are two species, not three: C. acicula and L. trochiformis. The text refers to one species and two subspecies of Creseis.

In line 71, where says “L. inflata (A. d’Orbigny, 1834) and H. inflatus “, must be a comma, instead of “and”.

In line 92, where says: “…oceanic region, and 3) these studies have been mainly conducted in..”, remove the words “these studies have” to make the sentence grammatically consistent.

Experimental design

One of the gaps addressed in the work and is correctly mentioned in the introduction states that previous studies are biased because they were conducted in certain regions of the Gulf, or in warm months, and that therefore there are data gaps regarding ocean waters or cold months.

CORRECTIONS:

In the results sections, in line 275 authors say that “It is important to note that heteropods of the genus Atlanta were not fully evaluated in this study because of the lack of species-level determination, which limited the assessment of heteropod richness”. This is more than a simple anottation to take in count in this work. Heteropods of the Atlanta genus are the most abundant within their group (in all ocean regions, the presence of others like Firoloida or Pterotrachea is insignificant compared with Atlanta abundances), so not having estimated their richness may wrongly influenced the design and results of the study. Please, consider add Atlanta specimens or restate the objectives of the article. In fact, in the discussion and conclusions the role of heteropods is not mentioned anywhere so, perhaps, heteropods should be out of this work.

In line 278, authors say that “The density of each species differed depending on the sampling depth. For example, the highest density values were found at 10 m depth… “ Do you refer to density of each species or density of total species? In fig. 5a seems to be a total species graph.

Validity of the findings

The discussion mainly refers to the richness and diversity of holoplankton mollusks by depth, in relation to the presence of the thermocline, and its relationship with ocean circulation patterns and the absence or presence of chlorophyll. Finally, the most abundant species are mentioned (H. inflatus, C. conica, and L. trochiformis).

The most relevant conclusion that fills the gap in knowledge is: “Temperature emerged as the primary physical factor affecting organism density values at the first two sampling levels, followed by dissolved oxygen and chlorophyll-a fluorescence, which may be related to adequate food availability (phytoplankton) for holoplanktonic mollusks in the euphotic zone”.

CORRECTIONS:

In relation to the species of the study, the authors analize the total abundances, and not species by depths (that I note as the most important variable and the key of this work). I think it is necessary to analyze the presence of dominant species according to samplings depth and contribute and discuss something about those results.

·

Basic reporting

In general, this article is well written and easy to follow. The study seeks to assess pteropod and heteropod populations in the Campeche Canyon, in the southern Gulf of Mexico, and how they relate to oceanographic variables. The rationale for performing the study is laid out comprehensively. There were some minor errors (mainly grammatical), which are listed below.

Some minor grammatical and/or referential errors:
• Lines 48 and 59-60: At a few places in the manuscript, but not throughout, pteropods are incorrectly referred to as heteropods. While both groups are in the class Gastropoda, Pteropods are in the subclass Heterobranchia and heteropods are in the subclass Caenogastropoda. Please make sure the correct group names are used throughout.
o Burridge et al., 2017 contains a phylogenetic description of pteropods vs heteropods if you would like a reference. (https://doi.org/10.1016/j.pocean.2016.10.001)
• Line 52: Did you mean to use “the highest” or “higher”? Zooplankton are not only attractive prey for top-trophic-level predators, but also mid-trophic-level predators. The source cited (Moreno-Alcántara et al., 2017) says that as well.
• Lines 81-86: The content of this sentence is good, but it’s long and doesn’t have a consistent grammatical structure. When making a list, each listed item should have the same grammar (for example: 1) noun, 2) noun, and 3) noun). The extra information about freshwater input could be a second sentence. As it is, the sentence is a bit too wordy.
• Lines 138 and 422: The name of the ship should be italicized (“R/V Justo Sierra”).
• Line 140: I don’t understand “separated every 14 nautical miles 120”. Is the “120” supposed to be there? Or is something else supposed to come after?
• Line 142: If you mean the sensors were calibrated before and after the cruise, then it should be “pre- and post-cruise” instead of “pre-post cruise”.
• Lines 143-144: To improve clarity, it should be “…from the sea surface to close to the bottom (5 m above the sea floor)” or something similar.
• Lines 148-149: The word “of” isn’t needed (“75 cm diameter, 250 cm total length”).
• Line 157: No comma should be used between “organisms” and “until”.
• Line 158: The word “continually” implies the lids were changed all the time. Could you please be more specific about the frequency?
• Line 171: I assume the unit is the number of individuals per 100 cubic meters. That unit of measure is not as common as meters or liters so please spell it out the first time it’s used.
• Line 225: Perhaps instead of “not shown”, you could refer the reader to the raw data, which contains the salinity data?
• Line 328: It should be “is” instead of “being”.
• Line 340: There should be “were” between “diversity” and “found” (“…diversity were found…”)
• Line 477: The link to the article doesn’t work.
• Lines 575 and 109: The year for the citation doesn’t match.
• Line 579 and 113: The name is hyphenated in the citation, but not in the reference.
• Lines 583 and 108: The names don’t quite match between the citation and the reference.
• Line 603 and 122: The name is hyphenated in the citation, but not in the reference.

Content organization could use improvement:
• Paragraphs, lines 59-80: These paragraphs feel like a random assortment of species and facts. Perhaps they could be reorganized to be easier to digest? The 3 categories you list in the following paragraph (lines 81-88) could be a good way to reorganize the facts: taxonomic composition, shallow coastal regions, and summer months. Another organization system would be fine too as long as it makes sense.

Experimental design

The sampling grid for this study was thorough and they processed a daunting number of specimens in order to support their findings. The methodologies used were suitably described that others could replicate the work.

I couldn’t find any mention of the time of day at which the zooplankton samples were collected. Were they all at night, during the day, or a mix of both? If the collection included both daytime and nighttime samples, was the time of collection factored into the analyses? Please add this information.

Validity of the findings

Overall, the findings are not unreasonable. However, I think a little more work could be done to improve the support for the findings and to enhance the discussion. Besides those broader areas, some minor improvements could be made to enhance the readability of the figures and the raw data files.

A) I’m surprised that there’s no investigation of the statistical significance of the relationships identified between the species and environmental variables tested. The conclusions made about those relationships would be better supported with tests of significance, such as regression analysis. For examples of similar studies that included statistical significance, please see Burridge et al., 2017 (https://doi.org/10.1016/j.pocean.2016.10.001) and Gallego et al., 2014 (https://doi.org/10.1017/S0954102013000795), among others.

B) In lines 368-371 of the discussion, it says that this study’s results agree with those found in continental shelf regions influenced by freshwater inputs. However, in lines 83-85 of the introduction, it says that the southern gulf has “a marked difference in terms of the composition of organisms in the oceanic region” because of the freshwater inputs. It appears that the introduction is contradicting the discussion. Perhaps you could expand further in the discussion on why the Campeche Canyon region would have similar organism density values to areas with freshwater inputs, contrary to your expectations in the introduction?

C) In lines 409-411, it states that “Temperature emerged as the primary physical factor affecting organism density values at the first two sampling levels, followed by dissolved oxygen and chlorophyll-a fluorescence”. That sentiment is similarly reflected in the discussion in lines 373-381. However, if you look at the loadings for PC1 in Table 2, the strength of the effect is actually highest for dissolved oxygen. The proper order of strengths would be dissolved oxygen > conservative temperature > absolute salinity > geostrophic velocity speed > fluorescence of chlorophyll-a. Could you please explain what analyses support temperature having the strongest effect or update your discussion to reflect the loadings?

D) The raw data files could use a little update to improve readability. In both files, units are missing for almost all columns. In “moluscos primer.xlsx” the column names are abbreviated. I suspect that most are supposed to be species names, but I cannot tell what they are supposed to be in all cases. Please write the full names. Also, column A has “stations” misspelled. In “parametros primer.xlsx”, the second tab, “Hoja2” seems unnecessary.

E) Also regarding the raw data, why are only 21 of the 48 stations included? It seems to be an arbitrary selection of stations. The figures appear to include data from stations that are absent from the raw data file so I assume those data are available.

F) Figures 3 and 4 contain gaps within the area of the station sampling grid. Could you please explain why that data is absent? I see that some, but not all, data is absent for the affected station in Figure 5 as well.

G) Not a requirement, but if it’s easy to add the T-T’ transect line to Figures 3 and 4 (like in Figure 1b), then it would be easier to put them into context when reviewing Figure 5.

H) Similarly for Figure 7, this is not a requirement, but it would be easier to read if the circles weren’t the same color as the background lines. If it’s not too much work to change the circle (density value) colors, that would improve readability.

I) The description for Figure 8 is a copy of the text for Figure 7. Please write an appropriate description for Figure 8.

J) For Figure 8a, please include a legend describing what the variables are (for example “T = conservative temperature”).

Additional comments

It’s clear that an impressive amount of work has gone into sampling, processing, and analyzing this study. I think it is a good study overall, but just has a few little knots to sort out before it is ready to publish. Once it is ready, I think it will be an important contribution to the understanding of holoplanktonic gastropods in the Gulf of Mexico.

---

## Round 0.2 · Major Revisions

The main issue raised by one of the reviewers is the data analysis, which seemed to be biased. I would like the author to focus on the analysis section and address all analytical and statistical comments in your next round of revision.

·

Basic reporting

In this second revision, all the corrections I had proposed have been made and the suggestions have been accepted by the authors, so I have nothing further to object to for its acceptance and publication.

Experimental design

In this second revision, all the corrections I had proposed have been made and the suggestions have been accepted by the authors, so I have nothing further to object to for its acceptance and publication.

Validity of the findings

In this second revision, all the corrections I had proposed have been made and the suggestions have been accepted by the authors, so I have nothing further to object to for its acceptance and publication.

Additional comments

In this second revision, all the corrections I had proposed have been made and the suggestions have been accepted by the authors, so I have nothing further to object to for its acceptance and publication.

·

Basic reporting

I appreciate the work that was done to improve the grammar and readability of the article. The paragraphs on Lines 76-111, especially, are much easier to follow.

Minor notes:
• The link on lines 608-611 still doesn’t work for me. If you’ve checked it, then perhaps it is a firewall issue on my end.
• Line 76 is missing a space after a period. “2009).In the southern…”
• Line 80 has a missing space and an extra space and comma. “acicula(Rang, 1828), , and…”

Experimental design

No comment.

Validity of the findings

Most of my comments have been addressed and I appreciate the work that has gone into doing so. The figures are overall much easier to follow. That being said, there are a several larger issues that remain outstanding.

A) Now that the authors have clarified that samples were taken both during the day and at night but that the time of sampling was not factored into the analyses, I am concerned that omission could be a form of bias affecting the results. It is well understood that zooplankton participate in diel vertical migration, remaining at depth during the day. Most of the transect and other sampling sites are in areas where the seafloor is 1000 m or deeper so it is reasonable to assume that many heteropods would be outside of the maximum net sampling depth (200 m) during the day. In my personal experience, there are noticeable differences in the number of zooplankton collected between day and night sampling. Ignoring a known source of bias in zooplankton collection makes the results unreliable. Please re-do the analyses to incorporate time of day as a variable.

B) I am glad to see that the authors have pursued statistical significance testing for temperature and that it supports their claims. However, temperature is only one of multiple factors included in the PCA plot. Please test the significance for each factor included, not just temperature.

C) I see that edits have been made to acknowledge that dissolved oxygen has a stronger effect than temperature per the PCA loadings. It seems that the discussion and conclusion nevertheless focus on temperature as a variable with only passing references to dissolved oxygen, latitude, and longitude—all of which had stronger effects than temperature. It is important to follow where the data leads and to not give the appearance of favoring an answer despite the results. I would like to see more than 1 sentence each discussing the 3 strongest variables (assuming latitude, longitude, dissolved oxygen, and temperature are still the most important after incorporating time of day as a variable). Once time of day is included in the analyses, please ensure that the strongest variables are all discussed.

D) I did not catch before that Figure 5 also does not have any statistical significance testing to support the conclusions made about depth. In order to make claims about differences between depths, it must be determined whether the values are statistically significantly different—appearances alone are not sufficient to determine a relationship. Please test the significance of the differences in density, species richness, and diversity. If it turns out that the differences are not significant, please update the discussion accordingly.

E) Regarding my comment about only 21 of 48 stations being included in the raw data file, I understand the authors’ explanation. However, even though only 24 stations were sampled for zooplankton, Figures 3 and 4 seem to use oceanographic data from all available stations. For example, Station 10 seems to be used in those figures, but it is absent from the raw data file. Please update the raw data file to include all stations used for the figures.

Additional comments

Some good improvements have been made since the last draft, but it is not quite ready yet. I am optimistic that the analysis changes recommended will improve the reliability of the conclusions.

---

## Round 0.3 · accepted · Accept

I want to congratulate the authors on their perseverance in making the article scientifically acceptable for publication in PeerJ. Although the comments were rigorous, the authors addressed all of them with proper responses.

I have no reservations about whether this article will be published in PeerJ. I am looking forward to seeing it as an open-access article.

·

Basic reporting

Regarding the link on line 627, I have determined that the issue must be my institution’s firewall. It still won’t load on the university’s network, but it works fine on other networks. My apologies for the confusion.

I have no further concerns about the basic reporting.

Experimental design

No further comments.

Validity of the findings

I appreciate the extensive work that the authors have done to address my concerns about the analyses. I am now satisfied with the methods used, how the results were discussed, and that the raw data is sufficiently available in the supplementary files.

Additional comments

All the changes I recommended have been implemented and I am happy to endorse this article for publication.